# Preparation of High Water-Soluble *Trichoderma* Co-Culture Metabolite Powder and Its Effects on Seedling Emergence Rate and Growth of Crops

**DOI:** 10.3390/jof9070767

**Published:** 2023-07-20

**Authors:** Lusheng Chen, Dazhi Hao, Kai Dou, Bo Lang, Xinhua Wang, Yaqian Li, Jie Chen

**Affiliations:** 1School of Agriculture and Biology, Shanghai Jiao Tong University, Shanghai 200240, China; cls199819@sjtu.edu.cn (L.C.); dz_574642689@foxmail.com (D.H.); dou_kaikai@126.com (K.D.); langbobo2@sjtu.edu.cn (B.L.); xhwang@sjtu.edu.cn (X.W.); yaqianli2008@163.com (Y.L.); 2State Key Laboratory of Microbial Metabolism, Shanghai Jiao Tong University, Shanghai 200240, China

**Keywords:** *Trichoderma*, β-cyclodextrin carrier, co-culture, growth promotion

## Abstract

*Trichoderma* spp. are widely used beneficial microbes in agricultural production; however, the improper carrier choice for *Trichoderma* agent preparation can alter the effectiveness of *Trichoderma* fungicides. In this study, the co-culture of four *Trichoderma* strains produced a large amount of free amino acids, with a content of 392.8414 ug/mL, and significantly improved the production level of γ-aminobutyric acid. A greenhouse experiment further showed that the co-culture of *Trichoderma* synergistically improved the female flower development and bacterial angular leaf spot resistance. The effects of ten kinds of carriers were compared in terms of water absorption and heat generation, as well as their effects on the seedling emergence rate and the plant growth promotion of maize, cucumber, and pakchoi cabbage. Each carrier was screened to mix with four strains of co-culture metabolites to prepare highly soluble and quality powders. The results showed that there were different effects of the carriers themselves and *Trichoderma* strain co-culture metabolite powder prepared with the carriers on seedling emergence rate and seedling growth. Β-cyclodextrin performed best in high solubility and low heat generation upon absorbing water and in easy drying in processing operations. *Trichoderma* strains co-culture metabolite powder with β-cyclodextrin as a carrier provided the most obvious promotion effects on seedling emergence rate and seedling growth. Therefore, β-cyclodextrin was determined to be an ideal carrier to prepare a highly water-soluble *Trichoderma* agent. Taken together, the study successfully developed a new type of highly soluble powder containing *Trichoderma* co-culture metabolites that is expected to benefit farming drip irrigation and spraying systems for the promotion of crop growth and disease control.

## 1. Introduction

*Trichoderma* is a commonly used microbial resource for the biological control of plant diseases and the promotion of crop growth [1]. In addition, it plays an important role in the promotion of nutrient utilization, yield, and quality in vegetables and food crops [2,3,4,5]. In the preparation process of biocontrol or plant growth promotion agents, the important materials for *Trichoderma* agent preparation are the types of physical carriers selected for absorbing spores and metabolites. The main *Trichoderma* carrier agents used are diatomite, bentonite, kaolin, talc powder, and other inert substances. They can not only provide mineral nutrients for *Trichoderma* and crops but also protect *Trichoderma* spore activity from environmental stress to extend the shelf life of *Trichoderma*-based products [6,7,8,9,10]. Generally, most carriers are insoluble and inert, which has a significant impact on the water solubility of *Trichoderma* agent products. The solubility of microbial agents becomes increasingly important when microbial agents are applied through drones or drip irrigation systems in modern agricultural production; however, in some cases, the excess carrier sediment formed in the use of microbial agents can block the pores of those farming irrigation systems [11,12]. For instance, *Trichoderma* wettable powders are often prone to precipitation after dilution with water due to poor water solubility and can thus block sieve nozzles, which seriously affects the quality of field application of microbial agents and the effects of disease prevention and growth promotion. Currently, commonly used highly water-soluble carriers, despite their good water solubility, generate high heat when exposed to water, affecting spore activity and leading to the loss of functional volatile substances produced by *Trichoderma* [13,14]. Therefore, the screening of optimal carriers with high water solubility and low heat production when diluted with water is of great importance to improve the effectiveness of the application of *Trichoderma* in modern farming systems, in addition to the basic requirements for carriers with a high adsorption capacity to microbial biomass and easy drying operations [15,16].

There have been few studies on the synergistic effects of interactions between fungi and physical carriers in microbial agents with the purpose of promoting crop growth. Early studies have shown that *Pseudomonas aeruginosa* RS-198 prepared with alginate, bentonite, and starch as carriers increased the biomass, soluble protein content, and chlorophyll content of cotton grown under saline soil conditions [17], and the combination of *Pseudomonas aeruginosa* DRB1 and *Trichoderma* harzianum CBF2 using talc as a carrier effectively controlled banana wilt and enhanced microbial viability [18]. However, to date, there has been a lack of research on the preparation of highly water-soluble *Trichoderma* metabolites powder agents made from the liquid fermentation broth of *Trichoderma* strain co-cultures and soluble physical carriers. Therefore, in this study, we compared the drying and heat production properties of ten different sources of carriers and the effects of the selected carriers individually combined with the co-culture of multiple *Trichoderma* strains (i.e., co-fermentation) on seed germination and growth of different crops, aiming to select suitable carriers for the preparation of highly water-soluble agents of *Trichoderma* strains’ co-culture metabolites and to separately evaluate their promoting effect on plant seedling growth.

## 2. Materials and Methods

### 2.1. Strains

The *Trichoderma* strains were provided by the *Trichoderma* Preservation Center of Shanghai Jiao Tong University (Table 1, Appendix A).

### 2.2. Culture Medium

Potato dextrose agar (PDA) medium (g/L): potato 200 g, glucose 20 g, agar 20 g. Distilled water was added to 1 L, and the pH value was neutral.

Potato dextrose (PD) medium (g/L): potato 200 g, glucose 20 g. Distilled water was added to 1 L, and the pH value was neutral.

Co-culture medium (g/L): corn meal 50 g/L, KH_2_PO_4_ 3.82 g/L, NaNO_3_ 1.42 g/L, (NH_4_)_2_SO_4_ 1.1 g/L, NaCl 1 g/L, MgSO_4_·7H_2_O 0.5 g/L, FeSO_4_·7H_2_O 0.0075 g/L, MnSO_4_ 0.0025 g/L, ZnSO_4_ 0.002 g/L, and pH neutral.

### 2.3. Crop Cultivars

Seeds of cucumber (*Cucumis sativus* L.) Shenqing No. 1 were purchased from Shanghai Funong Seed Industry Co., Ltd.

Seeds of cultivars of maize seed (*Zea mays* L.) Zheng Dan 958 were purchased from Henan Goldoctor Seed Co., Ltd.

Seeds of pakchoi cabbage (*Brassica chinensis*) Yipin No. 3 were purchased from Shouguang Xinxinran Horticulture Co., Ltd.

### 2.4. Microbial Carriers

Pregelatinized starch (FG) was purchased from Qufu Tianli Pharmaceutical Accessories Co., Ltd., Qufu, China; β-cyclodextrin (FG) was purchased from Mengzhou Huaxing Biological Chemical Co., Ltd., Mengzhou, China; Xiwang maltodextrin was purchased from Shandong Xiwang Sugar Co., Ltd. Shandong, China(named XW-maltodextrin); Liang gong maltodextrin was purchased from Mengzhou Golden Corn Co., Ltd. Mengzhou, China (named LG-maltodextrin); Dextrin (PG) was purchased from Liaoning Dongyuan Pharmaceutical Co., Ltd. Liaoning, China; Shandong Huayao Jiayi powder (TG) was purchased from Shandong Haoyao New Material Co., Ltd. Shandong, China (named SDHY Jiayi powder); No. 1 Weifang Guanxiang Jiayi powder and No. 2 Weifang Guanxiang Jiayi powder were purchased from Shandong Weifang Guanxiang Jiayi powder factory, Shandong, China (named No. 1 SDGX-Jiayi powder and No. 2 SDGX-Jiayi powder_2); Shandong Jinan Yande Jiayi powder and Shandong Jinan Yande Jiawei powder were purchased from Shandong Jinan Yande Biotechnology Co., Ltd. Shandong, China (named SDJN Jiayi powder, SDJN Jiawei powder). The physical properties of the carrier are listed in Appendix A.

### 2.5. Cultivation Matrix

Vegetable cultivation organic matrix (organic matter content 55.2%, N content 1.03%) was purchased from Danyang Maohe Organic Fertilizer Co., Ltd.

## 3. Methods

### 3.1. Preparation of Trichoderma Co-Culture Metabolic Liquid Powder

#### 3.1.1. Preparation of the Co-Culture Medium

To prepare the *Trichoderma* spore suspension, 5 mL of distilled water was added to a five-day *Trichoderma* colony culture at 28 °C. The spores were then scraped off with a sterile spatula, transferred into a 10 mL sterile centrifuge tube, and adjusted to a spore concentration of 1 × 10^8^ CFU/mL.

A 0.5 mL *Trichoderma* spore suspension was transferred to a 250 mL conical flask containing 100 mL PD medium, which was then grown at 28 °C in a shaking incubator at 200 rpm for 2 days.

The initial inoculation solution was prepared as a sequential inoculation with 0.33% (*v*/*v*) *Trichoderma* inoculant: the first was RW10569-1 for 8 h of incubation, the second was SBW10264-1 for 10 h, and the third was GDFS1009-1 and CM100Z4-1. All inoculated cultures were grown in a 50 L fermenter for 5 d. Fermentation parameters were designed along *Trichoderma* growth stages in a fermenter (Table 2). The statistics were as follows.

#### 3.1.2. Composition Analysis of the Metabolic Filtrate

For the determination of free amino acids, the sample was vortexed for 1 min and centrifuged at 8000 rpm at 4 °C for 10 min, and then the supernatant was taken for further analysis of each amino acid content. The analysis was conducted using an amino acid extraction kit (Leagene Biotechnology, Article No. tc2153), and the amino acid content in the fermentation filtrate was assayed by ninhydrin colorimetry. The type and content of amino acids in the fermentation broth were determined by an automatic amino acid analyzer (Hitachi L-8900, Japan).

The operation method was as follows: the co-culture broth after centrifugation was analyzed directly on the column; the sample volume on the column was determined according to the sensitivity of the automatic analyzer used. The determination was performed at pH 5 to 5.5 and 100 °C. The reaction proceeded for 10 to 15 min, and the generated purple substance was then colorimetrically determined at 570 nm. The yellow compound generated was colorimetrically determined at 440 nm. It usually takes only approximately 20–30 min to perform a full amino acid analysis. The final automatic calculation gives the exact type and content of amino acids.

#### 3.1.3. Determination of the Carrier Dissolution Heating Curve

The co-fermentation supernatant stored at 4 °C was mixed with pregelatinized starch, β-cyclodextrin, XW-maltodextrin, LG-maltodextrin, dextrin, SDHY-Jiayi powder, No. 1 WFGX-Jiayi powder_1, SDJN-Jiayi powder, No. 2 WFGX-Jiayi powder, and SDJN-Jiawei powder, used at a ratio of 2:3 (*w*/*w*). The temperature dynamic changes during the mixing process were monitored with a temperature recorder, the recording time interval was 1 s, and the temperature error was ±0.1 °C. The temperature change curve gradually decreased after the recorded temperature rose to the highest point.

#### 3.1.4. Determination of the Carrier Drying Water Loss Curve

The prepared metabolic liquid powder was evenly laid on a 9 cm surface dish and dried in an oven at 40 °C. The water loss mass was measured every 12 h, and the water loss curves of the metabolic liquid powder agent were made according to the kinds of carriers mixed with the metabolic liquid powder.

#### 3.1.5. Carrier Solubility Determination

The sample was dissolved in standard hard water containing calcium magnesium compound at 30 °C, turned upside down 15 times, allowed to stand for 5 min, and filtrated through a 75 μM test sieve. The filtrate residues remaining on the sieve were then quantitatively determined. To measure the solution stability, the solution was allowed to stand for 18 h and then filtered again with a 75 µm test sieve.The preheated standard hard water at 30 °C was put into a 250 mL graduated cylinder up to 2/3 volume, in which a certain amount of sample (the number of samples should be consistent with the recommended maximum concentration, no less than 3G) was added. Eventually, the whole volume in the cylinder was adjusted to scale by adding standard hard water. The cylinder was then allowed to stand for 30 s, turned upside down by hand 15 times, and reset. The interval time for reversing and resetting once was not allowed to exceed 2 s.The solution was allowed to stand in the cylinder for 5 min ± 30 s and poured into a 75 pm test sieve with constant weight. The filtrate was collected into a 500 mL beaker for the next test. The cylinder was washed five times with 20 mL of distilled water. All insoluble substances were transferred into the sieve, and the washing solution was discarded. The test sieve was dried at 60 °C to a constant weight and eventually weighed (accurate to 0.0001 g).The solution in the beaker was allowed to stand for 18 h and filtered through a 75 µm test sieve. The test sieve was washed with 100 mL distilled water, dried at 60 °C to constant weight, and weighed (accurate to 0.0001 g).

### 3.2. Pot and Field Experiments of Crop Growth Promotion and Development

For reconfirming the effects of Trichoderma strain co-culture on cucumber growth and development. In the greenhouse, individual strain PD monoculture filtrate, combined monoculture filtrates of four Trichoderma strains, and co-culture filtrate of four Trichoderma strains were applied to drench the cucumber seedling after transplant, 10 plants were randomly selected for each treatment with culture filtrate, and each plant was drenched with 100 mL (culture filtrate after being diluted 100 times). In the cucumber female flower stage, the plant growth and host resistance performance to common foliar and root diseases were investigated. The plant height, leaf number, and female flower number of cucumber were counted. The chlorophyll content was measured by the Plant Nutrition Tester (No.TYS-3N). The determination result of chlorophyll content is SPAD value, which is in direct proportion to chlorophyll. Therefore, SPAD value can indirectly represent chlorophyll value to evaluate plant health and growth status.

In the pot experiment, β-cyclodextrin, XW-maltodextrin, and LG-maltodextrin were each mixed with *Trichoderma* co-culture filtrate into different filtrate powders according to the ratios of the various components, as summarized in Table 3. The filtrate powder was diluted with water into filtrate powder solutions of 10 (T1) and 100 (T2) times based on the contents of the active ingredients. The 50 mL dissolved powder filtrate powder was poured into a pot horticulture substrate.

Five seeds of cucumber and maize and eight seeds of pakchoi cabbage were sown in each pot of the various treatments, with the horticulture growth substrate containing only powder carriers without any co-culture filtrate as a control, with five replicates for each treatment in pots. The seedling emergence rate was investigated daily from 3 to 10 days after sowing. All pots from each treatment were placed under the conditions of a 16 h light/8 h dark photoperiod and room temperature at 24~25 °C for 15 d.

### 3.3. Statistical Analysis

Each biological treatment was repeated at least three times. GraphPad Prism 9.0 software was used to make graphs, and the data were analyzed using SAS 9.4 software by the least significant difference (LSD) method to compare the significance between different treatments.

## 4. Results and Analysis

### 4.1. Metabolic Principal Component Analysis in Trichoderma Strain Co-Culture

The yield of free amino acids was significantly higher in co-cultures with *Trichoderma* strains than in monocultures of each strain alone (Figure 1). The production of free amino acids in co-cultures of strains CTCCSJ-A-CM100Z4-1, CTCCSJ-A-GDFS1009-1, CTCCSJ-W-RW10569-1, and CTCCSJ-W-SBW10264-1 was 392.414 µg/mL, which was 60.55%, 74.38%, 9.83%, and 36.68% higher than that from every single strain, respectively, indicating that the production of amino acids in the fermentation broth could be improved by the co-culture of *Trichoderma* strains. The most significant difference in amino acid content between co-culture and monoculture was γ-Aminobutyric acid (γ-ABA) (Table 4).

### 4.2. Effect of Trichoderma Filtrate Metabolism on Cucumber Growth

Different fermentation filtrates had a very significant effect on the growth of cucumber. The growth of cucumber was significantly promoted by irrigation with *Trichoderma* metabolite. In the greenhouse condition, the combination of monoculture broth was the most obvious positive effect on promoting growth (Figure 2), with the growth height reaching 170.5 ± 21.5a, which is significantly different from other treatments. Z4-1 has the largest number of leaves. The number of leaves treated by co-culture and monoculture of 10264-1 and 1009-1 was more than that of the control, but there is no significant difference. The photosynthesis rate was also higher than that of the control, but there were no significant differences. Interestingly, co-culture metabolites showed an obvious improvement of cucumber female flower development and leaf resistance to bacterial angular leaf spot as compared with mono-culture and combined mono-cultures (Table 5).

### 4.3. Preparation of Highly Water-Soluble Co-Culture Metabolite Agents

#### 4.3.1. Comparison of Thermogenic Characteristics of Carriers

To compare the differences in soluble carriers, including pregelatinized starch, β-cyclodextrin, XW-maltodextrin, LG-maltodextrin, dextrin, SDHY Jiayi powder, SDJN Jiayi powder, SDJN Jiawei powder, No. 1 WFGX Jiayi powder, and No. 2 WFGX Jiayi powder, each carrier was mixed with *Trichoderma* co-culture metabolites at a 3:2 ratio.

The heat production upon adsorption of co-culture metabolites by carriers varied significantly among carriers, including pregelatinized starch, β-cyclodextrin, XW-maltodextrin, LG-maltodextrin, dextrin, SDHY Jiayi powder, SDJN Jiayi powder, SDJN Jiawei powder, No. 1 WFGX Jiayi powder, and No. 2 WFGX Jiayi powder, which generated the highest temperatures of 30.85 °C, 28.35 °C, 28.7 °C, 26.85 °C, 28.45 °C, 32.35 °C, 45.0 °C, 27.15 °C, 43.4 °C, and 30.8 °C (Figure 3), respectively. No. 1 WFGX Jiayi powder and No. 2 WFGX Jiayi powder showed extremely high heating capability, while SDHY Jiayi powder, pregelatinized starch, and Jiawei powder were slightly febrile, and there was no obvious heating phenomenon observed in other carriers. The overheating of carriers upon adsorption of co-culture filtrate would result in the loss of volatile functional components of co-culture metabolites; therefore, No. 1 WFGX Jiayi powder and No. 2 WFGX Jiayi powder were not suitable as soluble carriers to absorb the microbial filtrate.

#### 4.3.2. Changes in the Drying Characteristics of Carriers upon Mixing with Co-Culture Filtrate

Drying efficiency determination: The co-culture filtrate and carrier were mixed into 100 g of an agent at a ratio of 2:3 (*w*:*w*), and the water loss curve was determined at 40 °C. It was revealed that there were significant differences in the water loss rate among carriers (Figure 4), in which the co-culture metabolite powders with pregelatinized starch, β-cyclodextrin, and dextrins dried very quickly, with 12 h being long enough to reach full dryness. Others such as SDJN Jiawei powder were completely dried in 36 h, and SDHY Jiayi powder and SDJN Jiayi powder were completely dried in 48 h; however, XW-maltodextrin powder and LG-maltodextrin powder were not completely dried, even in 60 h. Thus, the drying efficiencies of XW-maltodextrin and LG-maltodextrin were too low to serve as carriers for the preparation of soluble co-culture metabolite powder.

Solubility assay: It was suggested that carriers had different solubilities. Pregelatinized starch had the lowest solubility, with 86.86% insoluble material; β-cyclodextrins were slightly insoluble, with 24.52% insoluble materials (Table 6); XW-dextrin and maltodextrin were both soluble. All eight carriers showed stability in solubility, and there was no visible insoluble material precipitated after 18 h.

Based on a comprehensive analysis of varied traits among carriers, β-cyclodextrins, XW-maltodextrins, and LG-maltodextrins were demonstrated to be quality candidate carriers for the preparation of highly water-soluble metabolic powders.

### 4.4. Enhanced Seedling Emergence rate and Seedling Growth by Highly Water-Soluble Co-Culture Filtrate Powders

#### 4.4.1. Promoting Effect on Seedling Emergence Rate

The different effects of co-culture filtrate powders on the seedling emergence rate of three kinds of crops were revealed in pot experiments, and there was a clear dosage effect of the co-culture filtrate powders used on the seedling emergence rate. β-cyclodextrins used as carriers used in co-culture filtrate powder were generally superior to the other carriers (Table 7, Table 8 and Table 9).

No tested carriers had any obvious effects on seedling emergence of the three crops (Table 7), but co-culture filtrate powders prepared with the carriers significantly promoted seedling emergence relative to seedling emergence in the control. There were significant improvements (*p* < 0.05) in pakchoi cabbage seedling emergence caused by co-culture filtrate powders prepared with the three carriers, and further, seedling emergence rate by the 100-fold diluted solution of co-culture filtrate powder was superior to that of the 10-fold diluted solution. Compared with control, the seedling emergence rate of pakchoi cabbage increased by 600.09%, 700.09%, and 612.60% (*p* < 0.05) at 7 d after treatment with the 100-fold diluted solution of co-culture filtrate powder that was prepared with the carriers β-cyclodextrin, XW-maltodextrin, and LG-maltodextrin, but insignificant effects on seedling emergence rate or negative impacts were observed if a 10-fold diluted solution of co-culture filtrate powder was used. In comparison, the 10-fold diluted solution of powders prepared with the carrier β-cyclodextrin performed significantly better in the improved pakchoi seedling emergence rate than the other two carriers. It can be concluded that there was an obvious dosage effect of the co-culture filtrate powders prepared with different carriers on pakchoi cabbage seedling emergence.

Similarly, the co-culture filtrate powders prepared with β-cyclodextrin and cyclodextrin had no significant effect on the maize seedling emergence rate but exhibited a certain inhibitory effect on the seedling emergence speed (Table 8). The 10-fold diluted solutions with co-culture filtrate powders that were prepared with the three carriers significantly inhibited the maize seedling emergence rate and seedling emergence speed (*p* < 0.05); however, the inhibition could be ameliorated when the powder was diluted up to 100 times. In comparison, a 10-fold diluted solution with co-culture filtrate powder prepared with the carrier β-cyclodextrin had the least inhibitory effect on maize seedling emergence. Therefore, the co-culture filtrate powders based on the three carriers also had dosage effects on maize seedling emergence.

Furthermore, co-culture filtrate powders with the carriers β-cyclodextrin and LG-maltodextrin significantly inhibited the cucumber seedling emergence rate and seedling emergence speed, but the XW-maltodextrin carrier powder had no significant effect (Table 9). The 10-fold diluted solution of co-culture filtrate powder with the three carriers presented obvious inhibitory effects on the seedling emergence rate and seedling emergence speed of cucumber (*p* < 0.05); comparatively, β-cyclodextrin had the least inhibitory effect. Compared with a 10-fold diluted solution of the metabolite powder, the 100-fold diluted solution offered positive effects on the seedling emergence rate and seedling emergence speed of cucumber. Overall, the promotional effect on crop seedling emergence depended on the dosage effect, regardless of the kinds of carriers used in the co-culture filtrate powder preparation.

#### 4.4.2. Promotion Effect on Seedling Growth

The three carriers themselves had no significant effects on the plant height, root length, above-ground plant part fresh weight, or the root fresh weight of maize seedlings (Figure 5a–d).

The 10-fold diluted solution of the co-culture filtrate powder had a certain inhibitory effect on the growth of maize seedlings grown in pots, while β-cyclodextrin as the carrier had a slight inhibitory effect. At the 100-fold dilution level, co-culture metabolic powder had a significant growth-promoting effect on maize seedlings. Co-culture filtrate powder with β-cyclodextrin as the carrier had the strongest promoting effect on the plant height of maize seedlings; however, there was no significant distinction among the three carriers in other plant promotion traits. Moreover, the co-culture filtrate powder with β-cyclodextrin as a carrier increased the plant height, root length, plant fresh weight, and root fresh weight of maize seedlings by 49.32%, 54.61%, 59.68%, and 59.04% relative to the control, respectively. Therefore, the plant promotion effect depended upon the dosage effect.

Regarding the effect on cucumber growth, the carrier revealed no significant impacts on plant height, root length, above-ground fresh weight, or root fresh weight (Figure 5e–h). The 10-fold diluted solution of co-culture filtrates with W-maltodextrins and LG-maltodextrins as carriers had significant inhibitory effects on the growth of cucumber seedlings in pot cultivation, while β-cyclodextrin as a carrier had no significant effect. The 100-fold diluted solution of co-culture filtrate powder had a significant growth promotion effect on cucumber seedlings in pots, and LG-maltodextrin as a carrier exerted a significantly lower promotion effect on cucumber seedlings than the other two carriers. The co-culture filtrate powder based on β-cyclodextrin as a carrier increased the plant height, root length, fresh weight of shoots, and fresh weight of roots of cucumber in pot cultivation by 28.45%, 47.24%, 35.81%, and 39.26% relative to the control, respectively, and the promotion effect depended upon the dosage level.

Regarding the effect on pakchoi cabbage growth, the carrier had no significant effect on plant height or whole plant fresh weight (Figure 5i,j). The 10-fold diluted solution of co-culture filtrate powder with cephalodextrin and glucondextrin as carriers had a significant inhibitory effect on the growth of pakchoi cabbage seedlings; the germination rate was very low, and the growth status was poor, but β-cyclodextrin as a carrier had no significant effect on pakchoi cabbage seedling growth. A 100-fold diluted solution of co-culture filtrate powder with β-cyclodextrin as a carrier increased the plant height and whole plant fresh weight of pakchoi cabbage seedlings in pot cultivation by 31.00% and 33.60% relative to the control, respectively.

The plant height, root length, aboveground fresh weight, and underground fresh weight of maize were analyzed by principal component analysis (Table 10). The plant height, root length, underground fresh root weight, and aboveground fresh weight of cucumber were analyzed, and the plant height and fresh weight of cabbage were analyzed (Figure 6). The contribution rate of each component to the principal component is between 8.5% and 9.5%, and there is no significant difference. However, the comprehensive score of different treatments is calculated, and the highest score is G, (filtrate powder with β-cyclodextrin as the carrier is diluted 100 times according to the active ingredient), which is significantly higher than the remaining treatments.

## 5. Discussion

In this study, a group of amino acid-rich metabolites was obtained through the co-culture of four *Trichoderma* strains, in which the amino acids released from the co-culture process reached 392.8414 ug/mL, the most abundant of which was alanine, which was significantly higher than observed in cultures of single strains. The most significantly different amino acid in content compared to the monoculture of single strains was γ-aminobutyric acid. Beyond amino acids, the four strains of *Trichoderma* co-culture filtrates are also rich in a range of proteins, carbohydrates, organic acids, and secondary metabolites (unpublished); thus, multiple strains in co-culture can provide plenty of nutrition for seedling emergence and crop growth. Greenhouse experiments showed that co-culture filtrates can increase the number of leaves relative to control, and more importantly, the co-culture was able to simulate female flower development and disease resistance to bacterial diseases which were extremely significant for the safety and high production of cucumber.

The above results demonstrated that the co-culture technique with multiple *Trichoderma* strains can significantly increase the level of amino acid production by *Trichoderma* rather than a single strain in monoculture. An early study conducted by Qiong Wu (2018) [19] also showed that the co-culture of *Trichoderma asperellum* with *Bacillus amyloliquefaciens* significantly increased the amino acid content, while in this study, we were the first to construct a co-culture system of multiple *Trichoderma* species or strains that were also able to improve the production of some crucial amino acids important to plant growth. Co-culture techniques have become quite common, and relevant studies have shown that the co-culture of *Trichoderma* can increase the production of metabolites, such as cellulose mold [20,21]. Studies have shown that [22] the co-culture of sg3403 and *Bacillus subtilis* 22 improves the secondary metabolites of antagonistic fungi. Compared to the 18 amino acids essential for plant growth, co-culture of the four strains produced 21 amino acids, of which 16 were the same as those essential for plant growth. Amino acids have been shown to promote seed germination and growth [23], and the γ-aminobutyric acid (GABA) produced by soaking seeds can promote the germination and growth of white clovers in saline environments [24].

The important quality indicators of modern biofertilizers or biopesticides depend on not only the strains used but also the kinds of microbial carriers used. In consideration of the sensitivity of living microbial cells and their volatile metabolites and the safety of processing microbes and secondary metabolites themselves under extreme stress conditions, the carrier commonly requires less heat production during processing, easy drying, and high solubility. Moreover, the carrier itself is better if it has a certain plant growth-promoting effect. In our study, the optimal carrier for the adsorption of *Trichoderma* multistrain co-culture filtrate was confirmed to be β-cyclodextrin. It was found that the seedling emergence of cucumber seeds was more sensitive to β-cyclodextrin and cereal dextrin than the other two crops tested. In contrast, maize and pakchoi cabbage seedling emergences were less sensitive to the three carriers, which demonstrated that the effect of either biofertilizer or biopesticide on plant growth depends upon the comprehensive relationship between bioagent carriers, strains, and the plant species they affect. Based on our results, importantly, the effect of the carrier itself on the germination of plant seeds needs to be considered first, while the type of carrier used is more flexible in design formulation. It has been shown that sludge ash containing elements similar to soil was a good choice as a carrier for increasing the germination of lentil seeds [25]; however, cyclodextrin was reported to impair wheat and maize seed germination [26]. In the evaluation of highly water-soluble metabolite powders prepared from each of the three carriers, it was suggested that β-cyclodextrin was suitable to prepare a water-soluble co-culture filtrate powder, for instance, a 100-fold diluted solution yielded significant promotion effects on maize, cucumber, and pakchoi cabbage, but the plant promotion effect depended on significant dosage effects. As *Trichoderma* metabolites generally contain gliotoxin, viridin, and its derivatives (dihydrogen viridin, 6-pentyl-a-pyrone and other secondary metabolites) [27], these secondary metabolites not only result in antagonistic effects on pathogenic fungi but also generate varied effects on plant growth. The effects of *Trichoderma* metabolites on plant growth depend on a dosage effect [28]. It has been demonstrated that the above secondary metabolites have inhibitory or toxic effects on seed germination and plant growth based on the secondary metabolite content *Trichoderma* produces at high concentrations; in contrast, at low concentrations, they act as growth regulators and promote plant growth [29,30,31]. In our case, it was inferred that 100-fold dilution of the water-soluble *Trichoderma* metabolite powder instead of 10-fold dilution was able to form an optimal concentration of *Trichoderma* secondary metabolites for the promotion of seed germination and seedling growth, which was fully consistent with the results of previous studies.

At present, most of the microbial pesticides or fertilizers used domestically and abroad have water solubility carriers that are not ideal. In application, the precipitation of insoluble carriers often occurs and blocks the apparatus, negatively influencing the application effect. It was found that although the water solubility of Gayi powder was the most ideal, there was obvious heat production after absorbing water, and the preparation process easily caused the loss of volatile functional metabolites, which was also unfavorable to spore activity. β-Cyclodextrin, although slightly precipitated in water, produced heat and dried quickly after absorbing water, facilitating a more efficient preparation.

It has been shown that β-cyclodextrin can provide a carbon source for *Trichoderma* and crop growth [32] and can also wrap metabolite molecules in a cavity [33], thus improving the thermal stability of the wrapped material [34]. β-Cyclodextrin as a carrier to prepare live microbial agents has already been found to be beneficial to the growth of biocontrol microbes and plants. HARMAN et al. [35] used β-cyclodextrin to make a powder of *Trichoderma* harzianum metabolites, which then revealed positive effects on plant growth and resistance against biotic and abiotic stresses. In addition, β-cyclodextrin is an oligomer of seven glucose molecules [36] and is commonly used as a protective agent to keep viable spores of *Trichoderma* safe when they are processed into agents and applied under natural stress conditions [37]. In addition, the metabolites can also be adsorbed in the cavity, which can increase the solubility, increase the compatibility between different substances, and play a slow-release role. Humic acid and other organic metabolites in the soil can combine with β-cyclodextrin and release slowly, which can maintain soil fertility for a long time and is of great significance to plant growth and development.

Above all, β-cyclodextrin was selected as an ideal carrier for the preparation of highly water-soluble *Trichoderma* metabolite powder, in which some synergistic effects occurred between the carrier and *Trichoderma* metabolites, underlining the stimulation of seed germination and crop growth. It was confirmed that β-cyclodextrin as a carrier upon absorbing the *Trichoderma* liquid co-culture filtrates did not release too much heat, consequently leading to reduced loss of volatile metabolites, which may be a reason why the combination of the co-culture filtrate powder with the β-cyclodextrin carrier presented an excellent performance in plant growth promotion; however, it is clearly suggested that the synergistic effects resulted from comprehensive factors and depended on the dosage effect, regardless of the carrier itself and the final co-culture powder action.

To conclude, the co-culture of Trichoderma multiple stains took advantage of the promotion of plant growth and development, as well as resistance to disease. Moreover, β-cyclodextrin-based *Trichoderma* strain co-culture filtrate powder was confirmed as a candidate quality biofertilizer or biofungide with high solubility in field applications for the promotion of crop seed germination, seedling growth, and the control of plant diseases.

## Figures and Tables

**Figure 1 jof-09-00767-f001:**
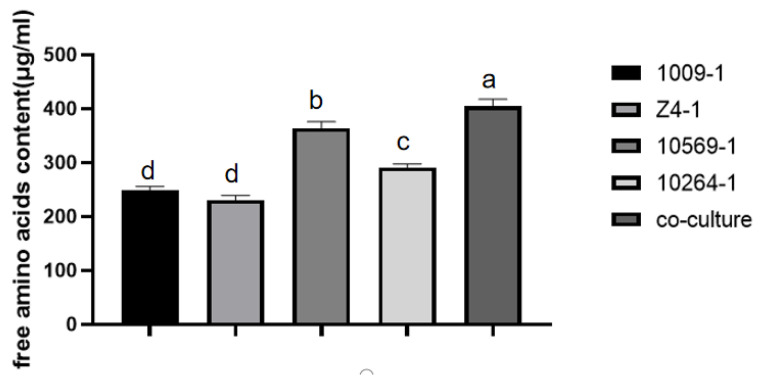
The yield of free amino acids in *Trichoderma* co-culture filtrate. 1009-1 denotes culture filtrate of *T. cf. kunmingense*; 10569-1 denotes culture filtrate of *T. cf. afroharzianum*; 10264-1 denotes culture filtrate of *T. cf. asperelloides*; Z4-1 denotes culture filtrate of *T*. *cf. asperelloides*; Co-culture denotes co-culture filtrate of the four *Trichoderma* strains above. Error bars represent the standard error. Bars with different letters represent a statistically significant difference from each other at the level of *p* < 0.05 based on the ANOVA.

**Figure 2 jof-09-00767-f002:**
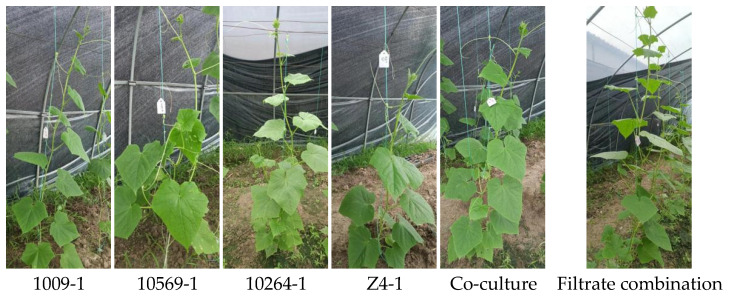
Field growth of cucumbers with different treatments. 1009-1 denotes culture filtrate of *T. cf. kunmingense*; 10569-1 denotes axenic culture filtrate of *T. cf. afroharzianum*; 10264-1 denotes culture filtrate of *T. cf. asperelloides*; Z4-1 denotes culture filtrate of *T. cf. asperelloides*; Co-culture denotes co-culture filtrate of the four *Trichoderma* strains above; Filtrate combination denotes mixing the monoculture filtrate of four *Trichoderma* strains.

**Figure 3 jof-09-00767-f003:**
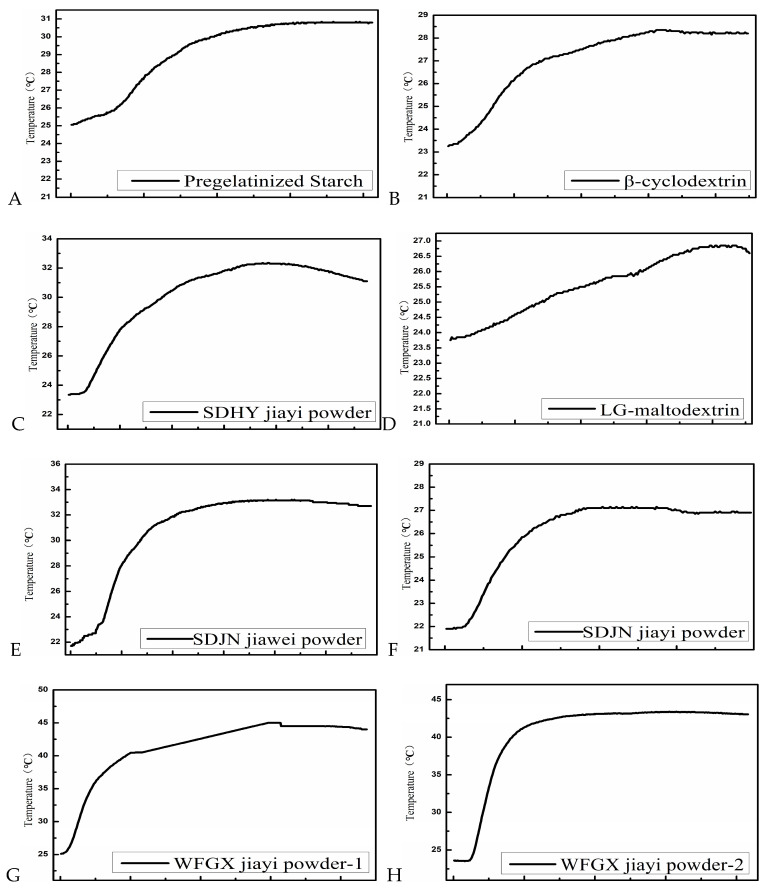
Heating curves of fermentation broth adsorbed by different carriers. (**A**) Pregelatinized Starch; (**B**) β-cyclodextrin; (**C**) SDHY Jiayi powder; (**D**) LG-maltodextrin; (**E**) SDJN Jiawei powder; (**F**) SDJN Jiayi powder; (**G**) No. 1 WFGX Jiayi powder; (**H**) No. 2 WFGX Jiayi powder; (**I**) XW-maltodextrin; (**J**) Dextrin. The temperature dynamic changes during the mixing process were monitored with a temperature recorder, the recording time interval was 1 s, and the temperature error was ±0.1 °C.

**Figure 4 jof-09-00767-f004:**
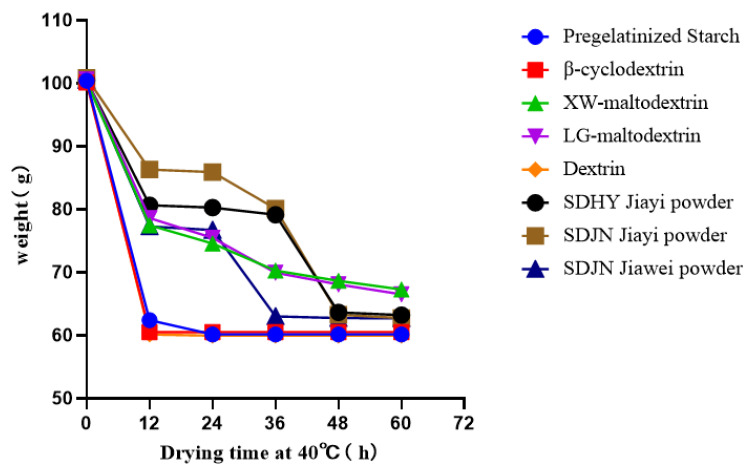
Water loss curves of the different carriers (Pregelatinized Starch, β-cyclodextrin, XW-maltodextrin, LG-maltodextrin, Dextrin, SDHY Jiayi powder, SDJN Jiawei powder, SDJN Jiayi powder). 40 g of each carrier absorbed with 60 g water was evenly placed into a petri-dish of 9 cm diameter and dried in an oven at 40 °C. The water loss mass was measured every 12 h.

**Figure 5 jof-09-00767-f005:**
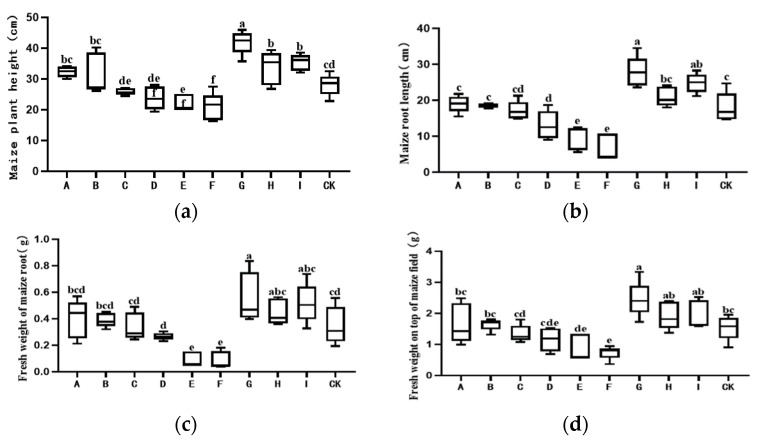
Effects of co-culture filtrate powder on plant growth. (**a**) The height of maize; (**b**) the root length of maize; (**c**) the fresh root weight of maize; (**d**) the aboveground fresh weight of maize; (**e**) the height of cucumber; (**f**) the root length of cucumber; (**g**) the aboveground fresh weight cucumber; (**h**) the fresh weight roots of cucumber; (**i**) the height of *Brassica chinensis*; (**j**) the fresh weight of *Brassica chinensis.* A β-cyclodextrin; B Xi Wang maltodextrin; C Liang Gong maltodextrin; D culture filtrate powder prepared with β-cyclodextrin as the carrier was diluted 10 times; E culture filtrate powder prepared with XiWang maltodextrin as the carrier was diluted 10 times; F culture filtrate powder prepared with Liang Gong maltodextrin as the carrier was diluted 10 times; G culture filtrate powder with β-cyclodextrin as the carrier was diluted 100 times; H culture filtrate powder prepared with XiWang maltodextrin as the carrier was diluted 100 times; I culture filtrate powder with Liang Gong maltodextrin as the carrier was diluted 100 times; CK sterile water. Five pots for each treatment were placed under the conditions of a 16 h light/8 h dark photoperiod and room temperature at 24~25 °C for 15 d. Error bars represent the standard error. Bars with different letters represent a statistically significant difference from each other at the level of *p* < 0.05 based on the ANOVA.

**Figure 6 jof-09-00767-f006:**
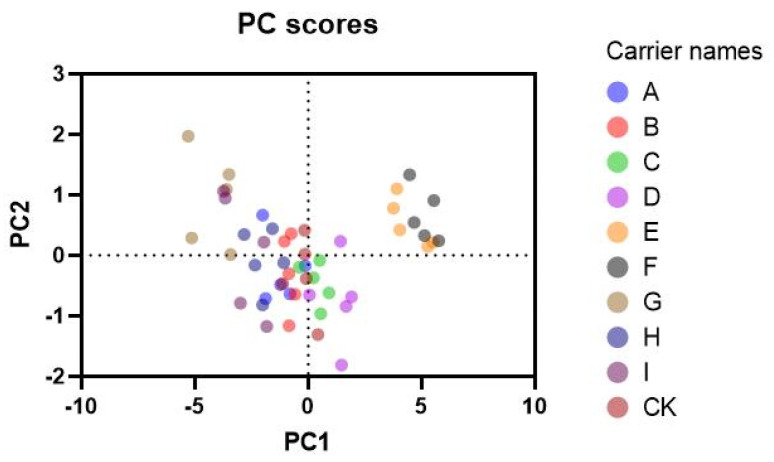
Principal component analysis of the effects of three metabolite powders on potted maize, cucumber, and *Brassica chinensis.* A−I represented the same treatments as those shown in Figure 5.

**Table 1 jof-09-00767-t001:** Information on *Trichoderma* strains tested.

Strain	Strain Number	Collection Location
*T. cf. asperelloides*	CTCCSJ-A-CM100Z4-1	Shanghai Chongming Wenzhou Mandarin bark
*T. cf. afroharzianum*	CTCCSJ-W-RW10569-1	Hainan—Wetland—soil
*T. cf. kunmingense*	CTCCSJ-A-GDFS1009-1	Guangdong—farmland—soil
*T. cf. asperelloides*	CTCCSJ-W-SBW10264-1	Hainan—Wetland—soil

**Table 2 jof-09-00767-t002:** Fermentation parameters of *Trichoderma* in different fermentation stages.

Fermentation Stages (h)	Temperature (°C)	Speed (rpm)	Ventilation Volume (vvm)	pH
A 0–24	28	200	1	5.3
B 24–72	28	180	0.8	2.7–4.0
C 72–96	28	160	0.6	3.2
D 96–120	28	160	0.6	5.0–8.0

A: Time for spore germination and mycelial growth; B: Starting time for conidia and chlamydospore formation; C: Time for conidia and chlamydospores produced in large quantities; D: Starting time of dissolution of the majority of hyphae.

**Table 3 jof-09-00767-t003:** Compositions and proportions of *Trichoderma* filtrate powders.

	Carrier (g)	Filtrate (g)	Water (mL)
T1-10	30	50	500
T2-100	30	5	500
CK	30	0	500

T1-10: The *Trichoderma* filtrate was diluted 10 times; T2-100: The *Trichoderma* filtrate was diluted 100 times before powder preparation; Carrier: there were ten different carriers: (a) Pregelatinized Starch; (b) β-cyclodextrin; (c) SDHY Jiayi powder; (d) LG-maltodextrin; (e) SDJN Jiawei powder; (f) SDJN Jiayi powder; (g) No. 1 WFGX Jiayi powder; (h) No. 2 WFGX Jiayi powder; (i) XW-maltodextrin; (j) Dextrin. The filtrate was *Trichoderma* spore-free solution through centrifugation at 8000 rpm for 10 mins.

**Table 4 jof-09-00767-t004:** Amino acid species and contents in co-cultures and monocultures of *Trichoderma* strains.

Amino Acid Species	Concentration of Amino Acids (μg/mL)				
	Co-Culture Filtrate	Z4-1 Filtrate	10569-1 Filtrate	1009-1 Filtrate	10264-1 Filtrate
P-Ser	11.8514b	7.32475d	29.3552a	2.34251e	9.19159c
Asp	1.71686e	7.17121a	2.35444d	5.8628025b	4.17685c
Thr	10.67645b	6.06562d	2.58427e	11.8959325a	7.31598c
Ser	1.24771c	9.82353a	1.42352c	1.2742275c	6.29698b
AspNH2	0.61149c	5.57618b	0.54141c	0c	7.89708a
Glu	5.47204c	48.31808b	11.56654c	44.141455b	71.70124a
Gly	24.02523b	14.66117c	45.23756a	9.8744325d	13.13442c
Ala	65.9561b	27.19946e	93.52672a	51.1219c	44.25904d
Val	29.59735b	18.23933c	37.11642a	13.08446d	10.56162e
Cys	2.13301c	0.74651d	5.32562b	2.365365c	6.38228a
Met	7.87293a	0.08405c	7.2444a	3.6592075b	0c
Ile	21.34237ab	9.14035b	23.76696a	9.7848475b	6.75262c
Leu	40.58132a	17.14349b	41.6599a	19.2698425b	10.64381c
Tyr	12.69384b	7.86794c	19.72159a	9.7938725c	7.76855c
Phe	20.6627a	6.33254c	17.27795b	9.7797625c	6.00279c
γ-ABA	86.04728a	18.8565b	8.31931c	9.3272575c	7.69485c
Orn	6.00145b	2.83885d	0.50519e	4.248885c	8.52352a
Lys	27.69712a	12.10821bc	1.04299c	13.22833bc	15.44427b
His	9.80399b	4.86493d	1.31043e	20.502775a	6.02619c
Arg	0e	0.91807d	1.43164c	3.1225975b	4.15478a
GluNH2	1.38565b	0c	1.61868b	0c	33.49787a
Trp	5.46513a	0c	4.74724b	0c	0c
Hypro	0	0	0	0	0
Pro	0	0	0	0	0
total	392.84142a	225.28077e	357.67798b	244.6804625d	287.42633c

1009-1 denotes culture filtrate of *T. cf. kunmingense*; 10569-1 denotes culture filtrate of *T. cf. afroharzianum*; 10264-1 denotes culture filtrate of *T. cf. asperelloides*; Z4-1 denotes culture filtrate of *T. cf. asperelloides*; Co-culture denotes co-culture filtrate of the four *Trichoderma* strains above. Error bars represent the standard error. Bars with different letters represent a statistically significant difference from each other at the level of *p* < 0.05 based on the ANOVA.

**Table 5 jof-09-00767-t005:** Effects of different *Trichoderma* filtrate on the growth of cucumbers.

Strains	Z4-1 Filtrate	1009-1 Filtrate	10264-1 Filtrate	10569-1 Filtrate	Combined ^a^ Filtrate	Co-Culture
Height of cucumber (cm)	102.8 ± 27.6b	158.5 ± 14.7a	154.1 ± 13.8a	148.5 ± 20.0a	170.5 ± 21.5a	154.7 ± 11.6a
Numbers of leaf	13.67 ± 2.08c	18.67 ± 3.06abc	16.33 ± 6.65bc	17.67 ± 3.06abc	23.67 ± 3.51a	21.33 ± 2.08ab
Diameter of Stem (cm)	1.44 ± 0.097a	1.39 ± 0.13ab	1.2 ± 0.094bc	1.21 ± 0.25bc	1.19 ± 0.19bc	1.18 ± 0.063c
Chlorophyll contents	43.7 ± 4.53	45.7 ± 4.76	42.9 ± 5.54	42.3 ± 2.68	42.4 ± 1.99	43.3 ± 3.65
Numbers of female flower	0	0	0	0	2	5
Bacteria leaf spot lesion ^b^	+	++	+	++	+	-

1009-1 denotes culture filtrate of *T. cf. kunmingense*; 10569-1 denotes culture filtrate of *T. cf. afroharzianum*; 10264-1 denotes axenic culture filtrate of *T. cf. asperelloides*; Z4-1 denotes axenic culture filtrate of *T. cf. asperelloides*; Co-culture denotes axenic co-culture filtrate of the four *Trichoderma* strains above. ^a^ combined filtrate was prepared by mixing monocultures filtrate of four *Trichoderma* strains; ^b^ + indicates the average number of bacterial angular leaf spot lesion in diseases leaf is less than 10, ++ indicates the average numbers of bacterial angular leaf spot lesion is over twenty; - indicates no lesion on plant leaves. Error bars represent the standard error. Bars with different letters represent a statistically significant difference from each other at the level of *p* < 0.05 based on the ANOVA.

**Table 6 jof-09-00767-t006:** Solubility and stability of different carriers.

	5 min	Insoluble Substances (g)	Proportion (%)	18 h
Pregelatinized Starch	Incomplete dissolution	2.1715 ± 0.0238a	86.86	Complete dissolution
β-cyclodextrin	Incomplete dissolution	0.6131 ± 0.0146c	24.52	Complete dissolution
XW-maltodextrin	Complete dissolution	0	0.00	Complete dissolution
LG-maltodextrin	Complete dissolution	0	0.00	Complete dissolution
Dextrin	Incomplete dissolution	1.742 ± 0.0116b	69.68	Complete dissolution
SDHY Jiayi powder	Complete dissolution	0	0.00	Complete dissolution
SDJN Jiayi powder	Complete dissolution	0	0.00	Complete dissolution
SDJN Jiawei powder	Complete dissolution	0	0.00	Complete dissolution

Insoluble substance means precipitant of the carrier in water in 20 min; Proportion (%) means the ratio of insoluble substances in carriers; 18 h means the time required for carrier complete dissolution. Error bars represent the standard error. Bars with different letters represent a statistically significant difference from each other at the level of *p* < 0.05 based on the ANOVA.

**Table 7 jof-09-00767-t007:** Effects of highly soluble filtrate powders on the seedling emergence rate of *Brassica chinensis* seeds in pots.

Carrier	Dilution Times	Seedling Emergence Rate (%)							
		3d	4d	5d	6d	7d	8d	9d	10d
β-cyclodextrin	-	0 ± 0j	11.11 ± 0hij	11.11 ± 0hij	11.11 ± 0hij	11.11 ± 0hij	11.11 ± 0hij	11.11 ± 0hij	11.11 ± 0hij
XW-maltodextrin	-	0 ± 0j	5.56 ± 6.42ij	5.56 ± 6.42ij	22.22 ± 0gh	22.22 ± 0gh	22.22 ± 0gh	22.22 ± 0gh	22.22 ± 0gh
LG-maltodextrin	-	0 ± 0j	11.11 ± 12.83hij	16.67 ± 6.42ghi	16.67 ± 6.42ghi	16.67 ± 6.42ghi	16.67 ± 6.42ghi	22.22 ± 12.83gh	22.22 ± 12.83gh
β-cyclodextrin	10	0 ± 0j	30.56 ± 16.67fg	38.89 ± 21.28ef	44.44 ± 28.69ef	47.22 ± 33.18e	50 ± 34.55e	50 ± 34.55e	50 ± 34.55e
XW-maltodextrin	10	0 ± 0j	0 ± 0j	0 ± 0j	0 ± 0j	5.56 ± 6.42ij	5.56 ± 6.42ij	5.56 ± 6.42ij	5.56 ± 6.42ij
LG-maltodextrin	10	0 ± 0j	0 ± 0j	0 ± 0j	0 ± 0j	0 ± 0j	0 ± 0j	0 ± 0j	0 ± 0j
β-cyclodextrin	100	0 ± 0j	66.67 ± 11.11d	75 ± 13.22abcd	75 ± 13.22abcd	77.78 ± 15.04abcd	77.78 ± 15.04abcd	77.78 ± 15.04abcd	77.78 ± 15.04abcd
XW-maltodextrin	100	2.78 ± 3.21ij	83.33 ± 7.86abc	84.72 ± 6.99ab	86.11 ± 7.17a	88.89 ± 4.54a	88.89 ± 4.54a	88.89 ± 4.54a	88.89 ± 4.54a
LG-maltodextrin	100	5.56 ± 6.42ij	68.06 ± 13.89cd	69.44 ± 11.56bcd	75 ± 13.22abcd	76.39 ± 11.45abcd	79.17 ± 11.45abcd	79.17 ± 11.45abcd	79.17 ± 11.45abcd
CK	-	0 ± 0j	11.11 ± 12.83hij	11.11 ± 12.83hij	11.11 ± 12.83hij	11.11 ± 12.83hij	11.11 ± 12.83hij	11.11 ± 12.83hij	11.11 ± 12.83hij

3–10 d was a different time to observe the Brassica Chinensis seedling emergence rate in the treatment of Trichoderma culture filtrate powders prepared with different carriers. Germination rate (%) = Number of germinated seeds/total number of seeds ×100%. Error bars represent the standard error. Bars with different letters represent a statistically significant difference from each other at the level of *p* < 0.05 based on the ANOVA.

**Table 8 jof-09-00767-t008:** Effects of highly soluble filtrate powders on the seedling emergence rate of maize seeds in pots.

Carrier	Dilution Times	Seedling Emergence Rate (%)							
		3d	4d	5d	6d	7d	8d	9d	10d
β-cyclodextrin	-	0 ± 0m	60 ± 23.09cde	60 ± 23.09cde	70 ± 11.55bcd	80 ± 23.09abc	80 ± 23.09abc	80 ± 23.09abc	80 ± 23.09abc
XW-maltodextrin	-	0 ± 0m	60 ± 23.09cde	60 ± 23.09cde	80 ± 0abc	80 ± 0abc	90 ± 11.55ab	90 ± 11.55ab	90 ± 11.55ab
LG-maltodextrin	-	0 ± 0m	30 ± 11.55ghijk	40 ± 0efghi	50 ± 11.55defg	60 ± 23.09cde	80 ± 23.09abc	80 ± 23.09abc	80 ± 23.09abc
β-cyclodextrin	10	0 ± 0m	30 ± 25.82ghijk	35 ± 34.16fghij	45 ± 34.16efgh	45 ± 34.16efgh	50 ± 34.64defg	50 ± 34.64defg	50 ± 34.64defg
XW-maltodextrin	10	0 ± 0m	5 ± 10lm	10 ± 11.55klm	15 ± 19.15jklm	15 ± 19.15jklm	15 ± 19.15jklm	20 ± 23.09ijklm	20 ± 23.09ijklm
LG-maltodextrin	10	0 ± 0m	0 ± 0m	0 ± 0m	5 ± 10lm	10 ± 11.55klm	10 ± 11.55klm	10 ± 11.55klm	10 ± 11.55klm
β-cyclodextrin	100	0 ± 0m	25 ± 25.17hijkl	85 ± 19.15ab	95 ± 10a	95 ± 10a	95 ± 10a	95 ± 10a	95 ± 10a
XW-maltodextrin	100	0 ± 0m	50 ± 25.82edfg	100 ± 0a	100 ± 0a	100 ± 0a	100 ± 0a	100 ± 0a	100 ± 0a
LG-maltodextrin	100	0 ± 0m	55 ± 19.15def	90 ± 11.55ab	95 ± 10a	95 ± 10a	95 ± 10a	95 ± 10a	95 ± 10a
CK	-	0 ± 0m	80 ± 0abc	90 ± 11.55ab	100 ± 0a	100 ± 0a	100 ± 0a	100 ± 0a	100 ± 0a

3–10 d was a different time to observe maize seedling emergence rate in the treatment of Trichoderma culture filtrate powders prepared with different carriers. Germination rate (%) = Number of germinated seeds/total number of seeds ×100%. Error bars represent the standard error. Bars with different letters represent a statistically significant difference from each other at the level of *p* < 0.05 based on the ANOVA.

**Table 9 jof-09-00767-t009:** Effects of highly soluble filtrate powders on the seedling emergence rate of cucumber seeds in pots.

Carrier	Dilution Times	Seedling Emergence Rate (%)							
		3d	4d	5d	6d	7d	8d	9d	10d
β-cyclodextrin	-	0 ± 0g	50 ± 11.55de	50 ± 11.55de	50 ± 11.55de	60 ± 0cde	60 ± 0cde	60 ± 0cde	60 ± 0cde
XW-maltodextrin	-	0 ± 0g	70 ± 34.64bcd	70 ± 34.64bcd	70 ± 34.64bcd	80 ± 23.09abc	80 ± 23.09abc	80 ± 23.09abc	80 ± 23.09abc
LG-maltodextrin	-	0 ± 0g	40 ± 23.09e	40 ± 23.09e	40 ± 23.09e	40 ± 23.09e	50 ± 34.64de	50 ± 34.64de	50 ± 34.64de
β-cyclodextrin	10	0 ± 0g	35 ± 34.16ef	40 ± 36.51e	40 ± 36.51e	50 ± 41.63de	55 ± 44.35cde	55 ± 44.35cde	55 ± 44.35cde
XW-maltodextrin	10	0 ± 0g	5 ± 10g	5 ± 10g	5 ± 10g	5 ± 10g	5 ± 10g	5 ± 10g	5 ± 10g
LG-maltodextrin	10	0 ± 0g	0 ± 0g	0 ± 0g	0 ± 0g	0 ± 0g	0 ± 0g	0 ± 0g	10 ± 11.55fg
β-cyclodextrin	100	0 ± 0g	55 ± 19.15cde	95 ± 10ab	95 ± 10ab	100 ± 0a	100 ± 0a	100 ± 0a	100 ± 0a
XW-maltodextrin	100	0 ± 0g	55 ± 25.17cde	95 ± 10ab	95 ± 10ab	100 ± 0a	100 ± 0a	100 ± 0a	100 ± 0a
LG-maltodextrin	100	0 ± 0g	55 ± 25.17cde	90 ± 11.55ab	100 ± 0a	100 ± 0a	100 ± 0a	100 ± 0a	100 ± 0a
CK	-	0 ± 0g	60 ± 0cde	70 ± 11.55bcd	90 ± 11.55ab	90 ± 11.55ab	90 ± 11.55ab	100 ± 0a	100 ± 0a

3–10 d was a different time to observe cucumber seedling emergence rate in the treatment of Trichoderma culture filtrate powders prepared with different carriers. Germination rate (%) = Number of germinated seeds/total number of seeds ×100%. Error bars represent the standard error. Bars with different letters represent a statistically significant difference from each other at the level of *p* < 0.05 based on the ANOVA.

**Table 10 jof-09-00767-t010:** Comprehensive score table of *PCA* analysis.

Carriers *	A	B	C	D	E	F	G	H	I	CK
PCA score	1.21	0.82	−0.36	−1.30	−4.49	−5.11	4.19	1.96	2.83	0.22

* A−I represented the same treatments as those shown in Figure 5.

## Data Availability

The data that support the findings of this study are available from the corresponding author, upon reasonable request.

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
