# Peer review of "Preparation of High Water-Soluble Trichoderma Co-Culture Metabolite Powder and Its Effects on Seedling Emergence Rate and Growth of Crops"

_jof, 2023, doi:10.3390/jof9070767_

Round 1

Reviewer 1 Report (Previous Reviewer 3)

Corrections have been made.

Figures are more clear now that the legend is more complete. 

I would suggest more comparaisons in the discussion, but it is okay as it is now for the direct objective of the study.

Reviewer 2 Report (Previous Reviewer 2)

Paper is now OK for publication. Congratulations.

This manuscript is a resubmission of an earlier submission. The following is a list of the peer review reports and author responses from that submission.

Round 1

Reviewer 1 Report

Review comments on “Preparation of high water soluble Trichoderma co-culture metabolite powder and effect on seed germination and growth of crops”

General comments

This is a research article about the application of Trichoderma to increase maize, cucumber and pakchoi cabbage seeds germination and plants growths. The study design is complete and several physiological indexes are applied to analyze the plant growth. The article can not be accepted with its current status, the authors should do a major revision before acceptance. And please see the comments below,

1.     Could you please give some explanation that why the physiological indexes used in the study;  

2.     how you control the high water soluble Trichoderma in the soil, as I know the contents will decrease when watering the plants in the soil; 

3.     it is better to set a blank control with fungi added but no plants, it is good to know the microbe growing condition without plant;

4.     Please redesign your figures, it is too much to be separated;

5.  Please add some more analysis methods to improve the study quality, such as PCA, RDA, etc., to analyze the physiological indexes;

6.  Please give a supplemental information for your strains of the Trichoderma spp.

Author Response

  Point1:Could you please give some explanation that why the physiological indexes used in the study;  

Answer: The co-culture agent quality depended upon application of optimal carrier since the  carrier has different characters in the heat release (Fig. 3) and drying efficacy (Fig. 4)

Point2: how you control the high water soluble Trichoderma in the soil, as I know the contents will decrease when watering the plants in the soil;

Answer:Our entire experiment was conducted for metabolites production from co-culture of Trichoderma strains, importantly, the metabolites agent was prepared by removing living spores  in fermentation broth through centrifugation. In addition ,high soluble co-culture metabolites agent was prepared by usingβ- cyclodextrin as carrier to absorb metabolites ,the carrier could be solved quickly in water.

Point3:it is better to set a blank control with fungi added but no plants, it is good to know the microbe growing condition without plant;

Answer:Our entire experiment was conducted using a Trichoderma fermentation broth without any living spores. The comparison we set up can already explain the results.

Point4:Please redesign your figures, it is too much to be separated;

Answer:Thank you. We have made changes according to your suggestions

Point5:Please add some more analysis methods to improve the study quality, such as PCA, RDA, etc., to analyze the physiological indexes;

Answer:Thank you. Now, according to your suggestion, the principal component analysis has been carried out on the growth indicators of Mazie, Cucumber and Brassica chinensis(Fig.6,Table 10)

Point5:Please give a supplemental information for your strains of the Trichoderma spp.

Answer:Thank you. The information and preservation number of the strain have been placed in Supplementary Material 1.

Reviewer 2 Report

Dear Corresponding Author
I checked your paper and I have some comments to improve your paper quality
1) Yoh have to correct so many spelling errors inside of the paper. For instance all of scientific name should be written as italics.
2) You need to imply the DNA characteristics of fungal isolates that you used into the paper (Trichoderma isolates). Accession numbers of DNA sequences and a phlogenetic tree is required.
3) In Figures 1, 3 and Thble 4 you have apply statistical analysis results.
With Best Regards
Reviewer

Author Response

Response to Reviewer 1 Comments

  1. Yoh have to correct so many spelling errors inside of the paper. For instance all of scientific name should be written as italics.

Answer:Thank you for your advice. Now these mistakes have been corrected.

  1. You need to imply the DNA characteristics of fungal isolates that you used into the paper (Trichoderma isolates). Accession numbers of DNA sequences and a phlogenetic tree is required.

Answer:Trichoderma asperelloides Z4-1(CGMCC NO.40245),T.harzianum RW10569(CGMCC NO.40246),T.asperellum GDFS1009(CGMCC NO.9512),T.asperellum SBW10264(CGMCC NO.22404) were all obtained from the China General Microbiological Culture Collection Center where the all tested strains are identified by ITS4/ITS5、rub2 and tef1.

  1. In Figures 1, 3 and Thble 4 you have apply statistical analysis results.

Answer:Thank you for your advice.

Reviewer 3 Report

Line 24: express in µg instead of ng

Line 31: excessive space to be removed

1.1.1:-Strain choice, can you please elaborate on why these 4 strains were chosen and not other strains (is it because of the crop choice, or was it more easily available?).

1.1.1-pH was natural: can you please specify the range or mention that it was not adjusted

-Line 110-125 can be presented as a table in supplementary material

-Table 2: precise range instead of pH natural

-Line 194 and 198: accurate to 00001g, is it 0.0001g or 1g?

-Line 203, missing space between “(T1)” and “and”, then “100” and “(T2)”

-Line 206: Five instead of 5

-Figure 1: convert units from ng to µg

-Figure 2: align graphs

-All figures: please add more details for figure captions (nb of replicates, method..) Readers must be able to understand content and description of the figures regardless of the article text

-Line 334: of high instead of ofhigh

-Figures 4-13: to be combined as panels of a single figure per plant species (one figure for maize roots and shoot characteristics, one for cucumber, one for Bok choy) or one for all three roots, one for three plant height…

-Figures 4-13: x axis carrier type needs to be defined in the figure caption for all figures instead of note A.

-line 392: according and not ACCording

-Line 476: to be rephrased, grammatical errors

-Discussion: Could you please elaborate on the effect of pH on the efficiency of microcarrier?

Authors need to discuss further the interaction between the chosen carrier and soil humic complexes. Also, what more fundamental hypothesis can they suggest for b-cyclodextrin as the best carrier regarding Trichoderma content and biochemical metabolite composition.

 -Further analysis of results between treatment and negative control should be highlighted.

Author Response

Response to Reviewer 3 Comments

  1. Line 24: express in µg instead of ng

Answer:Thank you.we have use µg instead of ng.

  1. Line 31: excessive space to be removed

Answer:Thank you.Now these mistakes have been corrected.

  1. 1.1:-Strain choice, can you please elaborate on why these 4 strains were chosen and not other strains (is it because of the crop choice, or was it more easily available?).

Answer:The selection of strains is based on previous studies in the laboratory, and the selection of carriers will be a key factor in the application of strains, so the strains used in this study are selected.

  1. 1.1-pH was natural: can you please specify the range or mention that it was not adjusted

Answer:Thank you.Now these mistakes have been corrected.

  1. -Line 110-125 can be presented as a table in supplementary material

Answer:Thank you.Now these mistakes have been corrected.Basic characteristics of the ten carriers are supplemented in Supplementary Material 2.

  1. -Table 2: precise range instead of pH natural

Answer:Thank you.Now these mistakes have been corrected.

  1. -Line 194 and 198: accurate to 00001g, is it 0.0001g or 1g?

Answer:Thank you.Now these mistakes have been corrected.it is 0.0001g,It means that the precision is high when weighing

  1. -Line 203, missing space between “(T1)” and “and”, then “100” and “(T2)”

Answer:Thank you.Now these mistakes have been corrected.

  1. -Line 206: Five instead of 5

Answer:Thank you.Now these mistakes have been corrected.

  1. -Figure 1: convert units from ng to µg

Answer:Thank you.Now these mistakes have been corrected.

  1. -Figure 2: align graphs

Answer:Thank you.Now these mistakes have been corrected.

  1. -All figures: please add more details for figure captions (nb of replicates, method..) Readers must be able to understand content and description of the figures regardless of the article text

Answer:Thank you.Now these mistakes have been corrected.

  1. -Line 334: of high instead of ofhigh

Answer:Thank you.Now these mistakes have been corrected.

  1. -Figures 4-13: to be combined as panels of a single figure per plant species (one figure for maize roots and shoot characteristics, one for cucumber, one for Bok choy) or one for all three roots, one for three plant height…

Answer:Thank you.Now these mistakes have been corrected.Integrate the pictures in Figure 5.

  1. -Figures 4-13: x axis carrier type needs to be defined in the figure caption for all figures instead of note A.

Answer:Thank you.Because there are many carriers, it seems too long in the title.

  1. -line 392: according and not ACCording

Answer:Thank you.Now these mistakes have been corrected.

  1. -Line 476: to be rephrased, grammatical errors

Answer:Thank you.Now these mistakes have been corrected.

  1. -Discussion: Could you please elaborate on the effect of pH on the efficiency of microcarrier?

Answer:Thank you.β-cyclodextrin contains 7 D-glucopyranose units, and ph will affect  Inclusion constant ofβ- cyclodextrin, and Hydrophobic interaction and hydrogen bond between β-cyclodextrins have influence. The main influence of Ph is the ionization balance of β-cyclodextrin will further affect the inclusion constant. Therefore, the metabolic solution needs to be within a certain ph range, and the ph value cannot be too low or too high.

  1. Authors need to discuss further the interaction between the chosen carrier and soil humic complexes. Also, what more fundamental hypothesis can they suggest for b-cyclodextrin as the best carrier regarding Trichoderma content and biochemical metabolite composition.

Answer:Thank you.It has been discussed in the article.

Round 2

Reviewer 1 Report

I think this version can be accepted, but the authors should use some references to support the answers next time. Good luck! 

Author Response

I think this version can be accepted, but the authors should use some references to support the answers next time. Good luck!

Answer:Thank you.Your suggestion is very useful. I will try to consult more literatures when replying in the future, and add relevant literatures when answering questions

Reviewer 2 Report

Dear Corresponding Author

Unfortunately you did not do my comments to improve your paper and therefore I cannot accept your paper to publication.

Regards

Author Response

Response to Reviewer 2 Comments

You need to imply the DNA characteristics of fungal isolates that you used into the paper (Trichoderma isolates). Accession numbers of DNA sequences and a phlogenetic tree is required.

Answer:Trichoderma asperelloides Z4-1(CGMCC NO.40245),T.harzianum RW10569(CGMCC NO.40246),T.asperellum GDFS1009(CGMCC NO.9512),T.asperellum SBW10264(CGMCC NO.22404) were all obtained from the China General Microbiological Culture Collection Center where the all tested strains are identified by ITS4/ITS5、tef1.

Point1:Unfortunately you did not do my comments to improve your paper and therefore I cannot accept your paper to publication.

Answer:I'm sorry, I haven't replied to you until now. Our laboratory is consistent that no one has submitted the genome on the NCBI website, so we just get Accession numbers now.The accession number of 1009-1 is OP799503;The accession number of 10264-1 is OP799504;The accession number of 10569-1 is OP799505;The accession number of z4-1 is OP799506.

Z4-1 was identified as Trichoderma asperelloide.1009-1was identified as Trichoderma . cf. Kunmingense.10569-1 was identified as Trichoderma. cf. Afroharzianum.10264 was identified as Trichoderma . cf. asperelloides
